# The Prevalence of Orthostatic Hypotension in Cancer Patients

**DOI:** 10.3390/cancers16081541

**Published:** 2024-04-18

**Authors:** Mateusz A. Iwański, Aldona Sokołowska, Andrzej Sokołowski, Roman Wojdyła, Katarzyna Styczkiewicz

**Affiliations:** 1Institute of Medical Sciences, College of Medical Sciences, University of Rzeszow, 1a Warzywna St., 35-310 Rzeszow, Poland; mati_747@interia.pl (M.A.I.); aldonasokolowskaa@gmail.com (A.S.); 2Collegium Humanum, Warsaw Management University, 133A Aleje Jerozolimskie St., 02-304 Warsaw, Poland; sokolows@uek.krakow.pl; 3New Medical Techniques Specialist Hospital of the Holy Family, Rudna Mała 600, 36-060 Rzeszow, Poland; romanwojdyla@gmail.com; 42nd Department of Cardiology and Cardiovascular Interventions, University Hospital, 2 Jakubowskiego St., 30-688 Krakow, Poland; 5Subcarpathian Oncological Centre, Frederic Chopin University Clinical Hospital, 2 Fryderyka Szopena St., 35-055 Rzeszow, Poland

**Keywords:** blood pressure, cancer, orthostatic hypotension

## Abstract

**Simple Summary:**

Cancer patients are characterised by a high prevalence of orthostatic hypotension (OH). Our study, for the first time, shows that the presence of cancer is a significant and independent predictor of OH, doubling the risk of OH compared to the noncancer population. The prevalence of OH differs depending on the type of cancer, being the highest among patients with lung cancer. The high prevalence of OH may be due to cancer itself, a consequence of oncologic therapies, or their side effects, which promote dehydration and low blood pressure. The presence of OH is considered an independent predictor of all-cause mortality in large clinical studies; therefore, its increased prevalence in cancer patients is of high clinical importance and requires particular attention. Screening and monitoring for the presence of OH is necessary and routine orthostatic challenge tests in the cancer population are recommended.

**Abstract:**

Background: Orthostatic hypotension (OH) is associated with a higher risk of mortality in the general population; however, it has not been studied in the cancer population. This study aimed to assess the prevalence of OH in cancer patients compared to that in the noncancer population. Methods: A total of 411 patients (mean age 63.5 ± 10.6 years) were recruited: patients with active cancer (*n* = 223) and patients hospitalised for other reasons, but without a cancer diagnosis (*n* = 188). Medical histories were collected and an orthostatic challenge test was performed. OH was defined as a blood pressure (BP) decrease upon standing of ≥20 mmHg for the systolic or ≥10 mmHg for the diastolic BP after 1 or 3 min; or a systolic BP decrease <90 mmHg. Results: The prevalence of OH in the subjects with cancer was significantly higher than in the subjects without cancer (28.7% vs. 16.5%, respectively, *p* = 0.003). OH was the most common in the lung cancer patients (57.5%). In a single-variable analysis, the predictors of OH were cancer presence, age ≥ 65 years, and body mass index (BMI) ≥ 30 kg/m^2^. In the multivariable model, the strongest independent predictor of OH was cancer status, which doubled the risk of OH, and BMI ≥ 30 kg/m^2^ and diabetes. Conclusions: Cancer patients are characterised by a high prevalence of OH. In this population, the recommendation of routine orthostatic challenge tests should be considered.

## 1. Introduction

Orthostatic hypotension (OH) is the result of the inability of the autonomic nervous system to control postural hemodynamic homeostasis caused by cardiovascular autonomic dysfunction [1,2]. It is recognised when cardiovascular adaptive mechanisms fail to compensate for the reduction in venous return that normally occurs on assuming the upright position and implies a persistent systolic/diastolic blood pressure (BP) decrease of at least 20/10 mm Hg upon standing. Cardiovascular autonomic dysfunction is one of the most poorly understood complications associated with cancer patients [3,4]. Several mechanisms have been proposed as explanations for the cause including OH, changes in heart rate (HR) frequency, and loss of BP and HR circadian variability [5,6].

OH has been widely studied in the general population and several associations with adverse events have been reported; therefore, it has been an established marker of a worse prognosis [1,2]. Patients with chronic diseases such as heart failure, diabetes, kidney dysfunction, and autoimmune disorders and those of an older age suffer from OH more frequently [7]. OH also often occurs alongside hypertension as a consequence of inappropriate antihypertensive drug use [7]. On the other hand, little is known about the prevalence of OH in cancer patients, although they are particularly susceptible and prone to dysregulation of BP and OH may also have a potential influence on prognosis in this population [8,9]. With this context, we decided to conduct this study to investigate the prevalence of OH in hospitalised active cancer patients compared to cancer-free patients.

## 2. Materials and Methods

Adult patients who were hospitalised in 2 centres, Subcarpathian Oncological Center of the University Clinical Hospital in Rzeszow and in the New Medical Techniques Specialist Hospital of the Holy Family in Rzeszow, were recruited for this study (Table 1). Given that age may significantly influence cardiovascular autonomic function, only subjects at age ≥ 40 years were recruited.

The study groups included patients with cancer actively treated with chemotherapy and/or radiation therapy (Table 1) and the non-oncological group, which included patients hospitalised for other reasons, but without cancer diagnosis. OH measurements in oncologic patients were taken 3 weeks after chemotherapy administration (just before the next cycle of chemotherapy) or throughout hospitalisation due to ongoing radiation therapy, or in patients with newly diagnosed cancer before oncological treatment was implemented. Patients with significant anaemia, requiring blood transfusion, or diarrhoea after recent surgical procedures or with acute life-threatening conditions were excluded. The following factors were subjected to analysis: the type of cancer, the severity of cancer disease, the presence of brain metastases along with concurrent diseases, and medications, especially those that may cause OH, such as alpha- and beta-blockers, nitrates, diuretics, calcium channel blockers, angiotensin-converting enzyme inhibitors, angiotensin II receptor blockers, and antiparkinsonian agents.

The data collected included demographic and medical history data, as well as measurements of sitting, supine, and standing BP on the orthostatic challenge test. BP measurements were performed according to the most recent standards of the Polish Hypertension Society [10] with a validated automatic upper arm device (Omron M3 Comfort, Omron Healthcare Co., Kyoto, Japan). OH was defined according to the criteria shown in Figure 1.

BP and HR were measured in the following order in each patient qualified for the project: three BP and HR measurements at intervals of 1–2 min in a sitting position, followed by a 5 min rest period; a double BP and HR measurement in the supine position after lying down for 5 min; a single BP and HR measurement 1 and 3 min after standing upright. For analysis, we calculated the mean of the last two measurements in a sitting position and the mean of double measurements while sitting. During standing, the following grades of accompanying symptoms were recorded: grade I: no dizziness; grade II: dizziness; grade III: syncope; and grade IV: prolonged disturbance of consciousness.

This study was approved by a local Bioethics Committee, no. 89/2022/B, and the patients gave their written informed consent to participate in this study.

### Statistical Analysis

For comparing two groups with respect to continuous variables, we used Student’s *t* test with separate variance estimators and a Mann–Whitney nonparametric test. For qualitative variables, the chi-squared test for independence was used, with test statistics calculated using the maximum likelihood approach. Risk factor analysis was performed using logistic regression. The multivariable model was obtained by backward step-wise approach. For all analyses, the significance level α has been set at 0.05. All calculations and graphs were prepared in STATISTICA 13 software.

## 3. Results

The current study included, in total, 411 patients (229 women and 182 men, mean age 63.5 ± 10.6 years). The distribution of the patients by hospital department is presented in Table 1. The cancer group consisted of 223 patients and a non-oncological group of 188 subjects. The data on the baseline demographic profile, type of cancer, and clinical characteristics of the enrolled patients are presented in Table 2 and Table 3.

In the patients with active cancer, 64 patients had OH (28.7%), while in the patients without a cancer diagnosis, the frequency of OH was significantly lower: 31 patients (16.5%)—*p* = 0.003. The cancer patients were statistically older, had a lower body mass index (BMI), a lower baseline HR, less frequent dyslipidaemia, and diabetes (Table 3). No statistically significant differences were observed with respect to sex, baseline systolic and diastolic BP values, the presence of other accompanying diseases, including hypertension, and drugs that can cause OH.

In the single-variable analysis, the predictors of OH were the presence of cancer, age ≥ 65 years, and BMI ≥ 30 kg/m^2^, which showed a negative correlation (Table 4). In the multivariable model, the strongest independent predictor of OH was cancer status; the patients with cancer were twice as prone to OH than the non-oncological patients (Table 4). The remaining independent predictors for OH in the multivariable analysis were BMI ≥ 30 kg/m^2^ (negative correlation) and diabetes, while an older age lost statistical significance. No significant differences were observed with respect to the sex of the patients, the presence of hypertension, and stroke.

The difference between the values of the systolic BP, diastolic BP, and HR in the supine position and after 1 and 3 min of standing was calculated. Delta BP was defined as the difference between the BP in an upright position and the BP in a supine position, calculated separately for the systolic and diastolic BP. Delta HR was defined as the difference between the HR in an upright position and the HR in a supine position. Figure 2 presents the different patterns of BP behaviour during the orthostatic challenge test—the systolic BP decreased while the diastolic BP increased. Some differences were noted with respect to the cancer status. The delta for the systolic BP both after 1 and 3 min of standing was greater in the cancer patients compared to the noncancer subjects (*p* = 0.02 and *p* = 0.04, respectively). The greatest fall in the systolic BP was observed after 1 min of tilting. The diastolic BP showed a different pattern: it increased on standing more in the noncancer patients, with a significant difference between the studied groups after 3 min (Figure 2). The HR increased in both groups during tilting: after 1 min of standing, it was greater in the cancer than in the noncancer subjects.

In the analysis of the different types of cancer, we noticed that OH was most common in the patients with lung cancer (*n* = 23/40; 57.5%), breast cancer (*n* = 6/16; 37.5%), male genitourinary cancers (*n* = 7/21; 33.3%), and head and neck cancers (*n* = 8/26; 30.8%). OH was less prevalent in the patients with gastrointestinal cancers (*n* = 10/52; 19.2%) and gynaecological cancers (*n* = 10/60; 16.7%)—Figure 3. OH was significantly more frequent in patients with lung cancer compared to other groups of malignancies, except breast cancer. OH did not depend on the stage of cancer progression (stage I–III versus stage IV, *p* = 0.19) and the presence of brain metastases (*p* = 0.52). The prevalence of OH was higher in the patients undergoing radiation therapy (*n* = 56) than those undergoing chemotherapy (*n* = 164)—24 (42.9%) vs. 39 (23.8%), *p =* 0.008, respectively.

## 4. Discussion

Our study, for the first time, shows that the presence of cancer is a significant and independent predictor of OH. Cancer presence doubles the risk of OH compared to in the noncancer population. In a multivariable analysis, the cancer status was a more powerful risk factor for OH than older age, BMI, and the presence of diabetes, widely known risk factors for OH.

OH may occur in cancer patients due to the cancer itself (tumour mass effect, secretory cancer activity, pro-inflammatory microenvironment), as a consequence of various oncologic therapies (chemotherapy, radiotherapy, surgery, analgesics), or due to the negative side effects of systemic treatment that lead to dehydration and favour low BP. On the other hand, cancer patients that are usually older and share similar cardiovascular risk factors with cardiac patients (i.e., lung cancer) may suffer from multiple age-related disorders like diabetes, hypertension, and chronic kidney disease, which, along with certain drugs, may further increase the risk of OH. In Figure 4, we propose several mechanisms as explanations for the increased prevalence of OH in cancer patients.

Autonomic nervous system dysfunction is a major problem that also affects cancer patients, with a prevalence reaching about 80% in patients with advanced cancer [1,2]. It has been described not only in patients with bronchogenic carcinoma but also in other malignancies including pancreatic, prostatic, breast, ovarian, and haematological cancers. Autonomic dysfunction is a complex syndrome involving the sympathetic and/or parasympathetic branches of the autonomic nervous system, which can, in turn, manifest as the dysfunction of organ systems (e.g., cardiovascular, gastrointestinal, or genitourinary). It is associated with chronic clinical disorders including diabetes mellitus, heart failure, and neurological diseases, with a negative impact on the affected patient’s prognosis. OH is one of the cardiovascular manifestations of an impaired autonomic nervous system.

Most of the available data on OH in cancer patients are only case reports [11,12,13,14,15]. Head and neck cancer patients undergoing radiation therapy have been shown to experience OH due to baroreflex insufficiency many years after treatment [16,17]. In ovarian cancer, chemotherapy with paclitaxel and carboplatin has been shown to negatively affect the cardiac sympathetic and parasympathetic nerves, which has been associated with an increased risk of OH [18]. In turn, it was observed that the combination of vinorelbine and cisplatin caused OH in some patients with metastatic breast cancer previously treated with anthracyclines and docetaxel [19]. In another study, it was discovered that patients with haematopoietic malignancies who undergo autologous stem cell transplantation or haematopoietic stem cell transplantation develop OH and often require treatment to control OH symptoms [20,21,22]. According to a prospective single-centre study conducted by Eriksen et al., orthostatic intolerance was a common problem in patients undergoing laparoscopic resection of colorectal cancer and was associated with delayed recovery [23]. Furthermore, OH has also been reported in patients with lung cancer after anatomical lung resections with thoracotomy, as well as lobectomy and segmentectomy assisted by thoracoscopy [24,25].

As information on the prevalence of OH in cancer patients is scarce, we decided to conduct this study to confirm, for the first time, that OH occurs more often in the oncologic population compared to subjects without cancer. Another interesting finding of our study was the surprisingly high prevalence of OH in patients with lung cancer. In our opinion, lung cancer patients are at an exceptional risk of cardiovascular autonomic dysfunction and related OH. Among the possible causes of the high prevalence of OH in this group of patients are the combination of various oncological treatment strategies (chemotherapy, radiation therapy, and surgery), the frequently advanced stage of disease, the presence of paraneoplastic syndromes, weight loss, and frequently overlapping cardiovascular diseases [3,26,27].

The frequency of OH in the general population ranges from 5% in patients <50 years of age to 30% in those over 70 years of age [28], which is consistent with the results of our study: the prevalence of OH in the noncancer patients over 40 years was approximately 17%. According to numerous clinical studies, OH is associated with adverse cardiovascular outcomes such as coronary artery disease, heart failure, atrial fibrillation, stroke, chronic kidney disease, and venous thromboembolism. The presence of OH is considered an independent predictor of all-cause mortality and noncardiac mortality in large studies; therefore, its increased prevalence in cancer patients is of high clinical importance and requires particular attention [29].

During the orthostatic challenge test, the physiological transition from the supine to the upright position is accompanied by a transient reduction in venous return with the pooling of intravascular blood volume in the lower extremities due to gravity. This leads to a decrease in the transient stroke volume and, consequently, a decrease in BP. In the normal scenario, the reflexes generated by the carotid sinus and aortic arch baroreceptors stimulate the sympathetic system and diminish the activity of the parasympathetic system. Due to the increased HR, cardiac contractility, and vascular tone, normal BP levels are restored [1]. In the case of the dysregulation of any of the phases of these mechanisms, an excessive BP decrease may occur, leading to OH. The orthostatic challenge test is usually associated with compensatory tachycardia, like in our study; in orthostatic hypotension of neurogenic origin, there is a minimal change in the HR despite significant hypotension.

Age is a significant risk factor for OH, and the incidence of OH is about 20% among people over 65 years [30]. The importance of age as a significant predictor of OH was also confirmed in our study. From this perspective, OH is a clinically relevant disorder, particularly for elderly patients who are more likely to fall, develop cardiovascular diseases, or cancer [31], and also significantly increases the risk of mortality of any cause [32,33,34].

Treatment-related OH has been observed with many drugs that affect the cardiovascular system and the central nervous system. Among these drugs are alpha- and beta-blockers, nitrates, diuretics, calcium channel blockers, angiotensin-converting enzyme inhibitors, and angiotensin II receptor blockers, benzodiazepines, antipsychotics, opioids, etc. [35]. In our study, we did not observe significant differences between the frequency of the use of these drugs between the cancer and noncancer patients, and they were not significant predictors of OH, maybe due to the relatively small population studied. We also did not observe the correlation between OH and hypertension in our study. This may be due to the more complex pathophysiology of OH in the cancer population. Of note, although hypertension is listed among the most frequent comorbidities in cancer patients [36], this diagnosis often refers to the patient’s medical history. In fact, due to episodes of hypotension that occur along with long-lasting oncological therapy, cancer patients often have their antihypertensive drugs discontinued.

OH can also be due to weight loss due to active cancer treatment or the advanced stages of the disease [37]. A study of 250 patients with a history of hypertension but without cancer showed that OH occurred in 9% of the patients with a BMI greater than 30 kg/m^2^. With the adoption of milder criteria for the diagnosis of OH, it was shown that the percentage of patients with a 10 mmHg decrease in the systolic BP was significantly higher among the obese (30.8%) than among the patients of normal weight (16%); *p* < 0.05) [38]. In our study, we found the opposite. In the single-factor analysis, we found that a higher BMI lowers the risk of OH. In the multifactor analysis, we found that a BMI above 30 kg/m^2^ is an independent predictor of the absence of OH. Our results can be attributed to cachexia, a debilitating, multifactorial, and often irreversible systemic syndrome that results in significant weight loss (primarily skeletal muscle and body fat). Around 50–80% of cancer patients suffer from cachexia, which contributes significantly to cancer-related mortality [39]. Therefore, it can be assumed that a higher BMI and obesity are a protective factor against OH in cancer patients.

The study by Van Hateren et al. showed that the prevalence of OH was higher in diabetic patients compared to healthy patients [40]. Our multifactor analysis confirmed that the patients with diabetes had a higher risk of OH, although the correlation was weaker than in the case of the patients with cancer. In another prospective study by Beretta et al., it was shown that patients with diabetes and OH were 2.7 times more likely to experience falls in hospital and 1.54 times more likely to die in hospital compared to patients without diabetes and OH [41].

In our opinion, by introducing routine measurements to detect OH in the cancer population, especially in lung cancer patients, we could identify patients at such risk, implement effective treatment, and prevent the consequences of OH. Furthermore, the early detection and prevention of OH consequences could improve the quality of life of cancer patients, which is already low due to other reasons. The presence of OH is considered an independent predictor of all-cause mortality in large clinical studies; therefore, its increased prevalence in cancer patients is of high clinical importance and requires particular attention with regard to its possible impact on cancer patients’ survival.

Cancer patients with confirmed OH, similarly to the general population, should be educated on their diagnosis and the goals of their treatment, which include improving excessive BP decrease and orthostatic symptoms after standing, without the worsening of already existing hypertension in some cases. Treatments include pharmacologic and nonpharmacologic methods. The latter are more commonly advised, and include the correction of reversible OH causes, increased fluid intake, sodium supplementation, the discontinuation of responsible drugs if possible, and the implementation of a specific training programme. Patients who do not respond adequately to nonpharmacologic interventions may be offered pharmacotherapy with fludrocortisone, midodrine, or pyridostigmine [1,2]. Telemedicine and mHealth systems that promote exercise can be considered helpful tools and are also beneficial in the cancer population [42].

The major limitations of our study include the relatively small study population and its heterogeneous characteristics (patients with various types of cancer on different oncologic therapies and at different disease stages). However, our efforts aimed to eliminate possible interference by including only stable middle-aged and old-age patients without relevant side effects of systemic therapy (vomiting, diarrhoea, dehydration). We have followed a strict measurement protocol; however, the OH challenge test was performed only once in each subject. The grading of the OH symptoms reported during the OH test could also be subjective; however, it did not impact on the OH diagnosis. We have not analysed in depth all the possible causes of OH and autonomic imbalance, which could include, i.e., the long-term effects of COVID-19 infection [43]. We have also not evaluated the possible role of additional tests, such as echocardiography and electrocardiography, and related scoring systems, which could play a potential role in the diagnosis and treatment of OH in the cancer population [44].

The role and high prevalence of OH in lung cancer patients deserves to be a topic of further study in this population, and the causes and consequences of this condition should be explored.

## 5. Conclusions

Cancer patients are characterised by a high prevalence of OH. The prevalence of OH differs with respect to the type of cancer, being the highest among patients with lung cancer. In the cancer population, the screening and monitoring of OH is necessary, and routine orthostatic challenge tests in this population are recommended. More research is needed to evaluate OH prevention methods and determine possible treatment interventions to avoid adverse events.

## Figures and Tables

**Figure 1 cancers-16-01541-f001:**
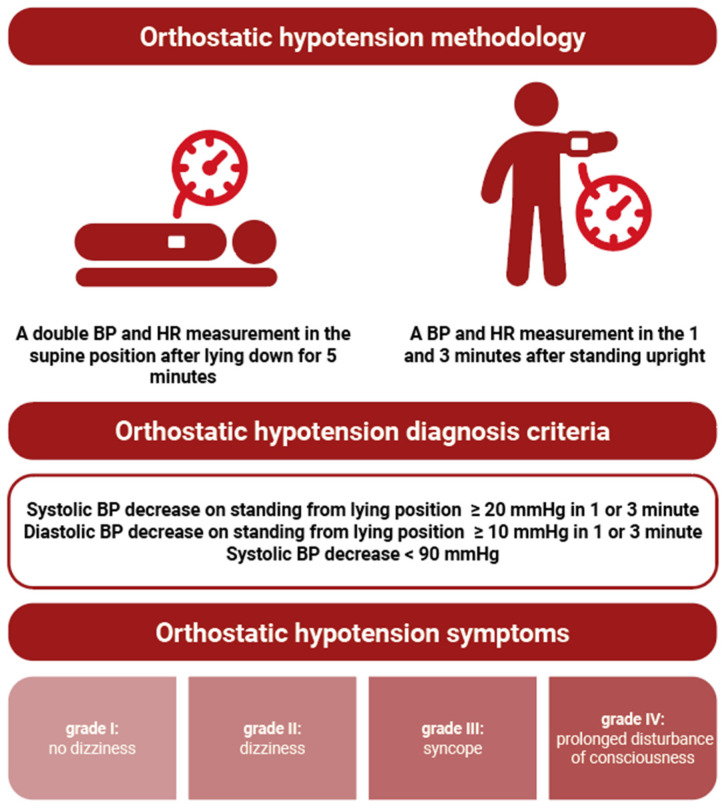
Outline of the methodology and diagnosis criteria of orthostatic hypotension. Abbreviations: BP, blood pressure; HR, heart rate.

**Figure 2 cancers-16-01541-f002:**
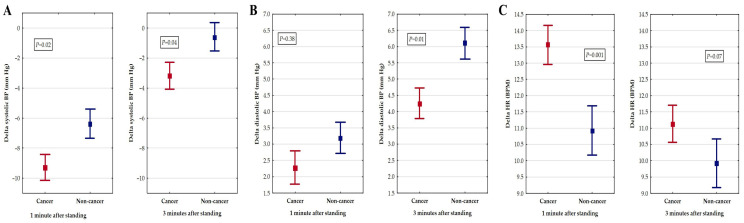
Blood pressure (BP) and heart rate (HR) changes in orthostatic challenge test according to cancer status. During tilting, a significant fall in systolic blood pressure is noted in cancer patients compared to noncancer subjects (**A**). Diastolic BP, however, rises upon standing, and after 3 min of testing, is significantly higher in non-oncological group than in cancer patients (**B**). HR is increased during tilting, and after 1 min, is significantly greater in cancer than noncancer patients (**C**). Abbreviations: Delta BP—difference between BP in upright position and BP in supine position, calculated separately for systolic BP and diastolic BP. Delta HR—difference between HR in upright position and HR in supine position. Red lines—cancer group, blue lines—non-cancer group.

**Figure 3 cancers-16-01541-f003:**
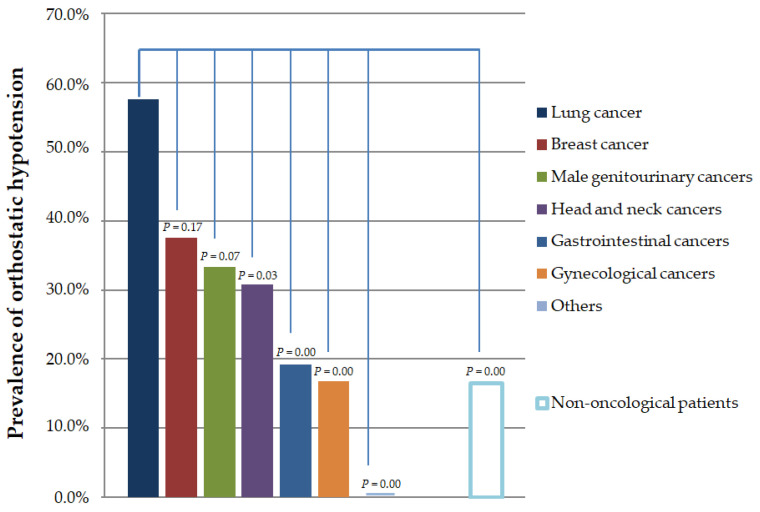
Prevalence of orthostatic hypotension according to the type of cancer. In lung cancer, orthostatic hypotension is significantly more prevalent compared to other cancer groups, except breast cancer.

**Figure 4 cancers-16-01541-f004:**
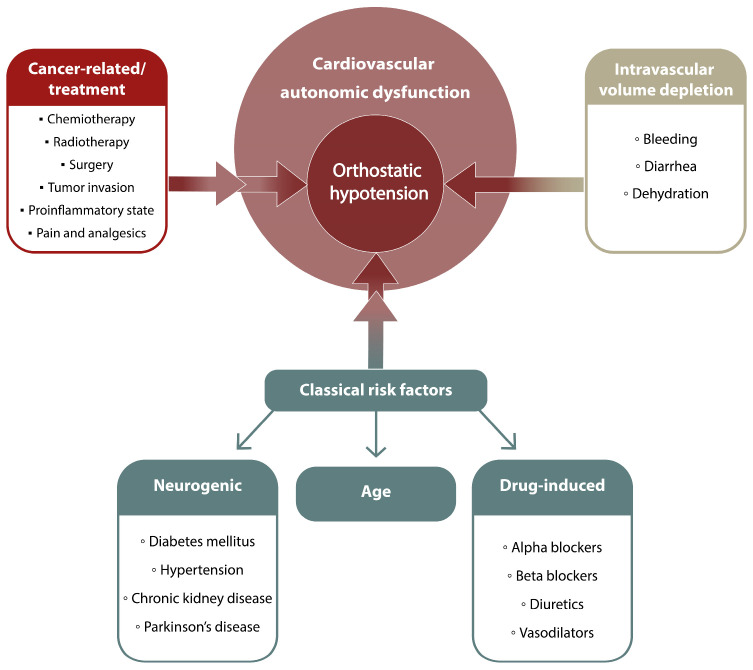
Main causes of orthostatic hypotension in cancer patients (description in the text).

**Table 1 cancers-16-01541-t001:** Distribution of patients according to hospital units.

Hospital Department	Number of Cases	%
Clinical Oncology	106	25.8
Oncological Gynecology	58	14.1
Radiotherapy	56	13.6
Neurology	108	26.3
Internal diseases	49	11.9
Dermatology	20	4.9
Rehabilitation	10	2.4
Urology	4	1.0
Total	411	100

**Table 2 cancers-16-01541-t002:** Distribution of patients according to type of cancer.

Type of Cancer	Number of Cases	%
Lung cancer	40	17.9
Breast cancer	16	7.2
Male genitourinary cancers	21	9.4
Head and neck cancers	26	11.7
Gastrointestinal cancers	52	23.3
Gynecological cancers	60	26.9
Others	8	3.6
Total	223	100

**Table 3 cancers-16-01541-t003:** Baseline demographic and clinical characteristics with respect to cancer status.

	Oncological (*n* = 223)	Non-Oncological (*n* = 188)	*p*-Value
Age in years, mean (SD)	65.0 (8.9)	61.5 (12.2)	0.01
Female sex, *n* (%)	116 (52.0)	113 (60.1)	0.09
BMI (kg/m^2^), mean (SD)	26.2 (5.7)	28.1 (5.0)	0.01
SBP (SD)	127.7 (16.6)	129.2 (18.4)	0.35
DBP (SD)	80.0 (10.4)	79.3 (9.7)	0.42
HR (SD)	69.4 (12.4)	78.4 (14.0)	0.001
Orthostatic hypotension, *n* (%)	64 (28.7)	31 (16.5)	0.003
Orthostatic hypotension symptoms:			0.00001
grade I, *n* (%)	198 (88.8)	187 (99.5)	
grade II, *n* (%)	25 (11.2)	1 (0.5)	
grade III, *n* (%)	0	0	
grade IV, *n* (%)	0	0	
Hypertension, *n* (%)	114 (51.1)	98 (52.1)	0.83
CAD, *n* (%)	22 (9.9)	21 (11.2)	0.66
Stroke, *n* (%)	5 (2.2)	9 (4.8)	0.15
Dyslipidaemia, *n* (%)	25 (11.2)	43 (22.9)	0.001
Diabetes, *n* (%)	38 (17.0)	48 (25.5)	0.03
CKD, *n* (%)	5 (2.2)	7 (3.7)	0.37
Thyroid diseases, *n* (%)	19 (8.5)	20 (10.6)	0.46
VTE, *n* (%)	15 (6.7)	7 (3.7)	0.17
PD, *n* (%)	1 (0.5)	3 (1.6)	0.23
Diuretics, *n* (%)	42 (18.8)	27 (14.4)	0.22
ACEi, *n* (%)	59 (26.5)	56 (29.8)	0.45
ARBs, *n* (%)	20 (9.0)	22 (11.7)	0.36
Beta-blockers, *n* (%)	82 (36.8)	60 (31.9)	0.30
Nitrates, *n* (%)	10 (4.5)	6 (3.2)	0.49
Alpha-blockers, *n* (%)	5 (2.2)	6 (3.2)	0.55
CCB, *n* (%)	34 (15.3)	38 (20.2)	0.18
Antiparkinsonian agents, *n* (%)	1 (0.5)	3 (1.6)	0.23

Abbreviations: ACEi, angiotensin-converting enzyme inhibitor; ARB, angiotensin II receptor blocker; BMI, body mass index; CAD, coronary artery disease; CCB, calcium channel blocker; CKD, chronic kidney disease; DBP, diastolic blood pressure; HR, heart rate; VTE, venous thromboembolism; PD, Parkinson’s disease; SBP, systolic blood pressure.

**Table 4 cancers-16-01541-t004:** Single and multivariable analysis of clinical predictors of orthostatic hypotension.

Variable	Single-Variable Analysis			Multivariable Analysis		
	OR	95% CI	*p*-Value	OR	95% CI	*p*-Value
Cancer	2.04	1.26–3.31	0.003	2.06	1.24–3.43	0.005
Age ≥ 65	2.45	1.44–4.18	0.0002	-	-	-
Male sex	1.31	0.83–2.08	0.24	-	-	-
BMI ≥ 30 kg/m^2^	0.47	0.26–0.82	0.007	0.40	0.22–0.72	0.002
Hypertension	1.47	0.92–2.35	0.10	-	-	-
Diabetes	1.61	0.94–2.74	0.07	1.90	1.06–3.40	0.03
Stroke	2.60	0.87–7.70	0.08	-	-	-

Abbreviations: BMI, body mass index; CI, confidence interval; OR, odds ratio.

## Data Availability

Data is contained within the article.

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
