# Peer review of "The Prevalence of Orthostatic Hypotension in Cancer Patients"

_cancers, 2024, doi:10.3390/cancers16081541_

Round 1

Reviewer 1 Report

Comments and Suggestions for Authors

1. In the selection of enrolled patients, the exclusion criteria only mention that "patients with severe anemia, diarrhea, recent surgery, or acute life-threatening conditions that require blood transfusion are excluded." Some patients with non neurogenic (such as burns, adrenal insufficiency, drug related, long-term bed rest or fever, etc.) and neurogenic (such as Parkinson's disease, multiple system atrophy, spinal cord injury, amyloidosis, etc.) that may cause OH are not included, and there is no description of whether they affect enrollment.

2. Analysis of sample size: Please evaluate how the sample size in this study was estimated and whether it is applicable to the statistical methods used in this study.

3. Please explain whether there were cases in this study where patients were unable to tolerate the experiment and left the group.

4. Clarifying the limitations of a study can increase its credibility and enhance its clinical guidance value. However, in the discussion of this study, the limitations and potential impact on the research results were not explicitly mentioned.Based on the limitations of this study, further directions for future research can be proposed.

5. Regarding future research directions, this study can further explore the impact of OH occurrence on the survival of cancer patients.

6. This study demonstrates for the first time that the presence of cancer is an important and independent predictor of OH, with significant clinical application value. In the discussion section, a paragraph can be reused to explain the important significance of this discovery in preventing OH in cancer patients, and specific useful preventive measures can be taken.

Reviewer 2 Report

Comments and Suggestions for Authors

-          Explain the significance or relevance of cardiovascular autonomic dysfunction in cancer and create a clear connection between orthostatic hypotension and chronic conditions.

-          Talk about orthostatic hypotension more in the introduction. Explain what happens to people when they suddenly stand. The information in the introduction is very minimal.

-          Table 1 would appear clearer if separated into two tables. This would allow you to give a title for each table, which clarifies that “hospital department” includes all patients (both onco- and non-oncological). Not separating the tables is a bit confusing.

-          Provide a brief rationale for choosing participants ≥40 years-old.

-          Explain the relation between heart rate and orthostatic hypotension.

-          Standardized criteria and details regarding how symptoms were assessed or recorded were not mentioned. This causes the grading system used by the authors to be subjective and inconsistent across patients. If no steps were taken to make the assessment more objective than subjective, please mention this in the limitations. If steps were taken please outline them in further detail in the methods section.

-          The discussion does not provide an explanation of why hypertension has non-significant correlation with OH meanwhile other studies demonstrate otherwise. Hypertension is one of the most common comorbidities associated with OH according to many researchers and according to your introduction. Some articles which could be helpful:

-          By Biaggioni et al. mentions the comorbidity of OH with hypertension https://doi.org/10.1093/ajh/hpy089

-          By Al-Ramahi et al. mentions the comorbidity of hypertension in breast cancer patients. https://doi.org/10.59049/2790-0231.1029

Reviewer 3 Report

Comments and Suggestions for Authors

I have reviewed the manuscript entitled 'The prevalence of orthostatic hypotension in cancer patients’.

The manuscript mentions an important issue in this frequent disease.The role of acute inflammatory diseases is very important in the change of blood pressure in patients with orthostatic hypotension. Please mention covid as an acute disease in the change of its autonomic functions. Please consider citing ‘Heart rate variability and cardiac autonomic functions in post-COVID period’There are important points which should be stated in the discussion section. The role of scoring systems are very important in order to diagnose and deal with orthostatic hypotension in cancer patients. There are a lot of scoring systems which are proved to have prognostic and diagnostic value in terms of ECG .  The authors should define scoring system and mention articles  ‘A simple formula to predict echocardiographic diastolic dysfunction-electrocardiographic diastolic index’ and ‘The significance of the morphology-voltage-P-wave duration (MVP) ECG score for prediction of in-hospital and long-term atrial fibrillation in ischemic stroke’.

What should be the treatment strategy of orthostatic hypotension in patients with cancer. Telemedicine and mHealth systems promoting exercise can directly effect prognosis in these patients. Please also mention this issue in a short section in the discussion citing ‘Lifestyle intervention using mobile technology and smart devices in patients with high cardiovascular risk: A pragmatic randomised clinical trial’ and ‘Telemedicine: Current Concepts and Future Perceptions’

Comments on the Quality of English Language

Minor editing of English language required

Reviewer 4 Report

Comments and Suggestions for Authors

The authors investigated the prevalence of orthostatic hypertension (OH) in 223 patients with cancer compared with 188 patients without cancer. However, I have some comments.

1)  What was the reproducibility of the methodology for the diagnosis of OH used in this study? Did the authors test OH twice in some patients? Were there any changes in the results?

2)  Was there any significant difference in the prevalence of OH among patients with chemotherapy, those with radiation therapy, and those with newly diagnosed cancer?

Reviewer 5 Report

Comments and Suggestions for Authors

This article by Ivanski et al looks into the prevalence of ortostatic hypotension in adult patients with different types of cancers.

Lines 43-44: “ Patients with chronic diseases such as hypertension, heart failure, diabetes, kidney dysfunction, autoimmune disorders and at older age suffer from OH more frequently”- OH in patients with hypertension may be a consequence of inapropriate antihypertensive medication, therefore hypertension should not be listed alongside with the other conditions, but seperately.

The introduction is too short

I noticed there was no standard application of the OH test: in some the test was performed 3 weeks after chemo or radiation therapy, in others before any therapy. This impairs your results.

I also noticed you only measured the blood pressure in the 1st and 3rd minute of standing. This excludes patients with delayed ortostatic hypotension. A typical standing test can be prolongued to up to 10 minutes. I think the extended version, that catches delayed symptoms should have been performed.

Why were only dizziness, syncope and prolonged disturbances of consciousness taken into consideration as OH symptoms? Why not nausea and fatigue? And what did you consider to be prolonged disturbances of consciousness?

Line 124: we should not confuse a standing test with a tilt test. What do you mean by active tilting?

Round 2

Reviewer 3 Report

Comments and Suggestions for Authors

Thank you for the required revisions. 

The paper is acceptable after the revision.

Author Response

Thank you for your vaulable comments and paper revision. 

Reviewer 4 Report

Comments and Suggestions for Authors

I have no further comments.

Author Response

Thank you for our paper revision and valuable comments. 

Reviewer 5 Report

Comments and Suggestions for Authors

The quality of the article has improved. One minor comment: line 264- please rephrase “cancer patients are frequently stopped with antihypertensive drugs”

Author Response

It was corrected

,,cancer patients often have their antihypertensive drugs discontinued''

Thank you for our paper revision and valuable comments.